# A helium-burning white dwarf binary as a supersoft X-ray source

J. Greiner[1✉], C. Maitra[1], F. Haberl[1], R. Willer[1], J. M. Burgess[1], N. Langer[2,3], J. Bodensteiner[4], D. A. H. Buckley[5,6,13], I. M. Monageng[5,13], A. Udalski[7], H. Ritter[8], K. Werner[9], P. Maggi[10], R. Jayaraman[11,12] & R. Vanderspek[11,12]

Type Ia supernovae are cosmic distance indicators[1,2], and the main source of iron in the Universe[3,4], but their formation paths are still debated. Several dozen supersoft X-ray sources, in which a white dwarf accretes hydrogen-rich matter from a non-degenerate donor star, have been observed[5] and suggested as Type Ia supernovae progenitors[6–9]. However, observational evidence for hydrogen, which is expected to be stripped off the donor star during the supernova explosion[10], is lacking. Helium-accreting white dwarfs, which would circumvent this problem, have been predicted for more than 30 years (refs. [7,11,12]), including their appearance as supersoft X-ray sources, but have so far escaped detection. Here we report a supersoft X-ray source with an accretion disk whose optical spectrum is completely dominated by helium, suggesting that the donor star is hydrogen-free. We interpret the luminous and supersoft X-rays as resulting from helium burning near the surface of the accreting white dwarf. The properties of our system provide evidence for extended pathways towards Chandrasekhar-mass explosions based on helium accretion, in particular for stable burning in white dwarfs at lower accretion rates than expected so far. This may allow us to recover the population of the sub-energetic so-called Type Iax supernovae, up to 30% of all Type Ia supernovae[13], within this scenario.

The X-ray source [HP99] 159 (ref. [14]) has been seen since the early 1990s with ROSAT, XMM-Newton (4XMM J052015.1–654426) and, recently, eROSITA (eRASSU J052015.3-654429) with a very soft spectrum (effective blackbody temperature of $kT = 45 \pm 3$ eV or $522 \pm 35$ kK; Fig. 1). Using the 1″-accurate XMM X-ray position, we identify [HP99] 159 with a 16-mag optical object at right ascension (RA) (2000.0) = 05 h 20 min 15.50 s, declination (dec.) (2000.0) = −65° 44′ 27.1″. An optical spectrum taken with the Robert Stobie Spectrograph (RSS) at the Southern African Large Telescope (SALT) shows a wealth of emission lines (Fig. 2), all shifted by the Large Magellanic Cloud (LMC) systemic velocity[15] of $262.2 \pm 3.4$ km s$^{-1}$, indicating that the source is indeed situated at LMC distance (50 kpc (ref. [16])). Thus, the X-ray fit leads to a high bolometric luminosity of $6.8^{+7.0}_{-3.5} \times 10^{36}$ erg s$^{-1}$. The corresponding blackbody radius is $3,700^{+3,900}_{-1,900}$ km, consistent with a white dwarf. This classifies [HP99] 159 as a bona fide supersoft X-ray source[5,17,18].

The optical spectrum is unique, in that it shows predominantly He I and He II emission lines (Fig. 2). There are no indications for Balmer lines (see insets in Fig. 2), no absorption lines typical for a main-sequence star and no indications for either C or O as seen in Wolf–Rayet stars. The only other emission lines we identify (Extended Data Fig. 2) are seven lines of N II (5,001.5, 5,666.6, 5,679.6, 6,482.0, 6,610.6 Å) and Si II (6,347.1, 6,371.4 Å). Although such lines are typically seen in AM

Canum Venaticorum (CVn) stars, several facts argue against such an interpretation (see Methods). We find no evidence in the extracted 2D long-slit spectrum of any extended nebulous emission. The strong continuum emission argues against an H II-like region of a (He-rich) planetary nebula.

High-resolution optical spectra taken at three epochs with the High Resolution Spectrograph (HRS) at SALT show a double-peaked profile of all lines (Fig. 3), thus demonstrating their origin in an accretion disk. With the theoretical maximum intensity of an accretion disk line profile coming from the area of about 0.95 of its maximum Doppler velocity, and assuming Keplerian rotation, we infer a projected velocity of the outer disk of $v_K \times \sin(i) \approx 60$ km s$^{-1}$. This suggests that the disk is seen at a low inclination angle, close to face-on. Notably, the He II lines have a similar profile. The full width at zero intensity in the He I lines suggests a maximum projected velocity of $120 \pm 10$ km s$^{-1}$, with that of the He II 4686 line clearly being different, about $200 \pm 20$ km s$^{-1}$.

The accretion disk is not only the origin of the emission lines but also of the ultraviolet–optical–near infrared continuum emission, as indicated by its luminosity and spectral slope; the accreting white dwarf and the donor are both hidden under this disk flux. Optical photometry shows periodic variations by a factor of 1.3, with little colour variation (see Extended Data Fig. 3). A Lomb–Scargle periodogram

[1]Max-Planck-Institut für extraterrestrische Physik, Garching, Germany. [2]Argelander-Institut für Astronomie, Universität Bonn, Bonn, Germany. [3]Max-Planck-Institut für Radioastronomie, Bonn, Germany. [4]ESO – European Organisation for Astronomical Research in the Southern Hemisphere, Garching, Germany. [5]South African Astronomical Observatory, Cape Town, South Africa. [6]Department of Physics, University of the Free State, Bloemfontein, South Africa. [7]Astronomical Observatory, University of Warsaw, Warsaw, Poland. [8]Max-Planck-Institut für Astrophysik, Garching, Germany. [9]Institut für Astronomie und Astrophysik, Kepler Center for Astro and Particle Physics, Universität Tübingen, Tübingen, Germany. [10]Université de Strasbourg, CNRS, Observatoire astronomique de Strasbourg, UMR 7550, Strasbourg, France. [11]Department of Physics, Massachusetts Institute of Technology, Cambridge, MA, USA. [12]Kavli Institute for Astrophysics and Space Research, Massachusetts Institute of Technology, Cambridge, MA, USA. [13]Present address: Department of Astronomy, University of Cape Town, Cape Town, South Africa. ✉e-mail: jcg@mpe.mpg.de

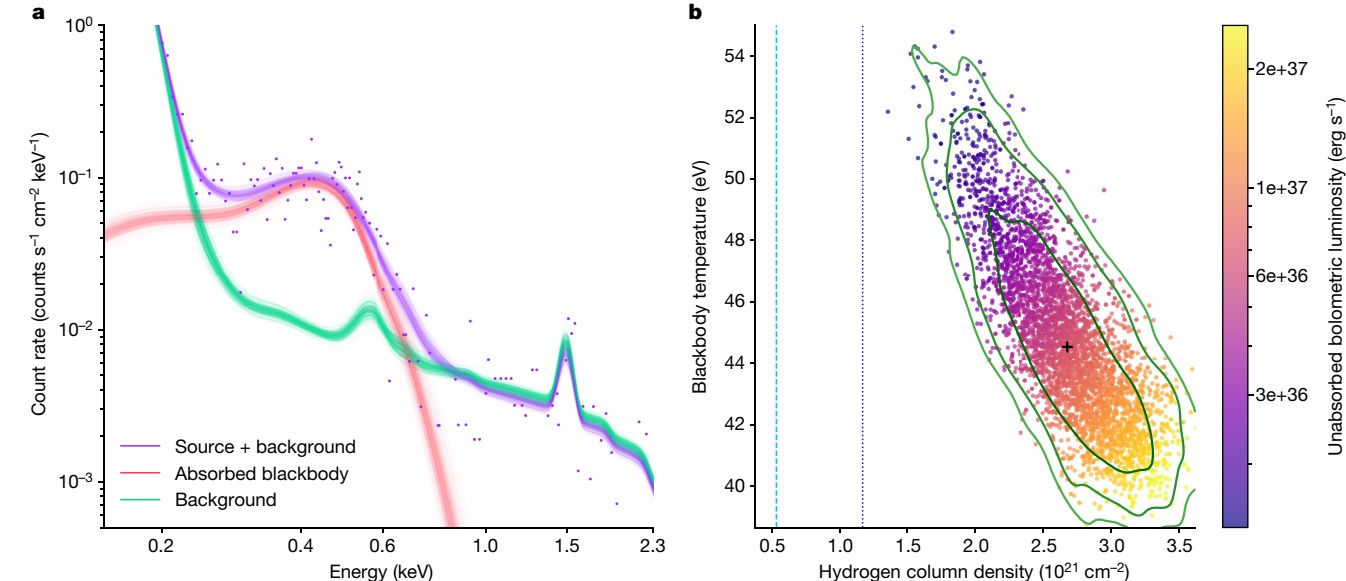

**Fig. 1 | X-ray temperature and luminosity constraints of [HP99] 159.**
**a**, A 3ML fit of an absorbed blackbody model to the XMM-Newton spectrum with a simultaneous fit of the background with linked parameters (see Methods) provides a good fit. Purple symbols and line show the source + background data and model, red the source only and green the total background. The individual background components are shown in Extended Data Fig. 4.

**b**, Posterior distribution in the temperature versus hydrogen column density plane of the spectral fit in **a**. Each dot represents one model realization. The colour coding represents the unabsorbed bolometric luminosity assuming a distance of 50 kpc. $1\sigma$, $2\sigma$ and $3\sigma$ confidence contours are overplotted in green. Vertical dashed lines mark the Galactic foreground absorption and the sum of Galactic and total LMC absorption.

shows the largest power at a period of 1.1635 days and a secondary lower-power peak at 2.327 days (Fig. 4). The folded light curve for this longer period has a lower variance and a clear odd–even asymmetry (Fig. 4d). Phase-resolved spectroscopy is certainly needed to firmly establish which one is the true orbital period.

The helium-dominated accretion disk has two consequences. First, the donor star must be in an evolutionary phase in which all the hydrogen is lost. An intriguing option is a helium star donor, with the nitrogen lines providing evidence for CNO-processed matter from the donor. Second, we interpret the high X-ray luminosity as resulting from steady

He burning in a shell on the white dwarf (accretor) surface. Similar to the steady H-shell burning in the canonical supersoft X-ray sources[19,20], models of accreting white dwarfs predict a narrow range of accretion rates, with a canonical value of about $10^{-6} M_{\odot}$ year$^{-1}$, at which He-shell burning is steady[7,8,11,12,21–23]. If the accretion rate is higher, the accreted material puffs up and forms an envelope around the white dwarf, which becomes similar to a red giant, probably leading to common envelope evolution. If the accretion rate is lower, burning in the accreted He layer is unstable, that is, first starting to oscillate and then leading to He-shell flashes that increase the luminosity temporarily by factors of

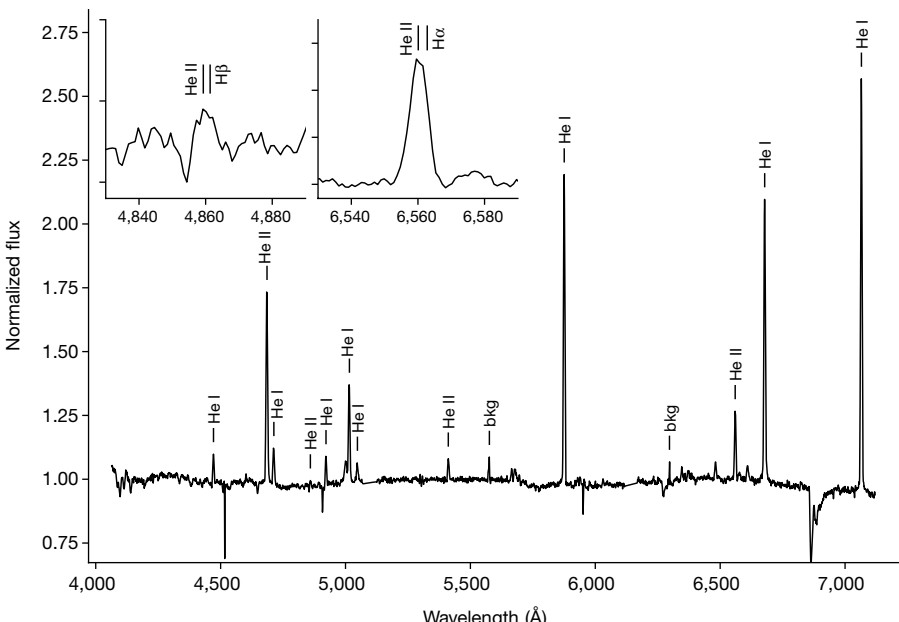

**Fig. 2 | Low-resolution optical spectrum of [HP99] 159.** Flux-normalized optical spectrum taken with the SALT/RSS spectrograph on 14 August 2020 at 03:44 UT (mid-time of 1,200 s exposure), with the main emission lines labelled ('bkg' labels residuals of removing sky lines). The three absorption features apart from the B-band are also a result of residuals of sky-line removal. The insets demonstrate that the 4,860-Å and 6,560-Å lines result from He II and not hydrogen.

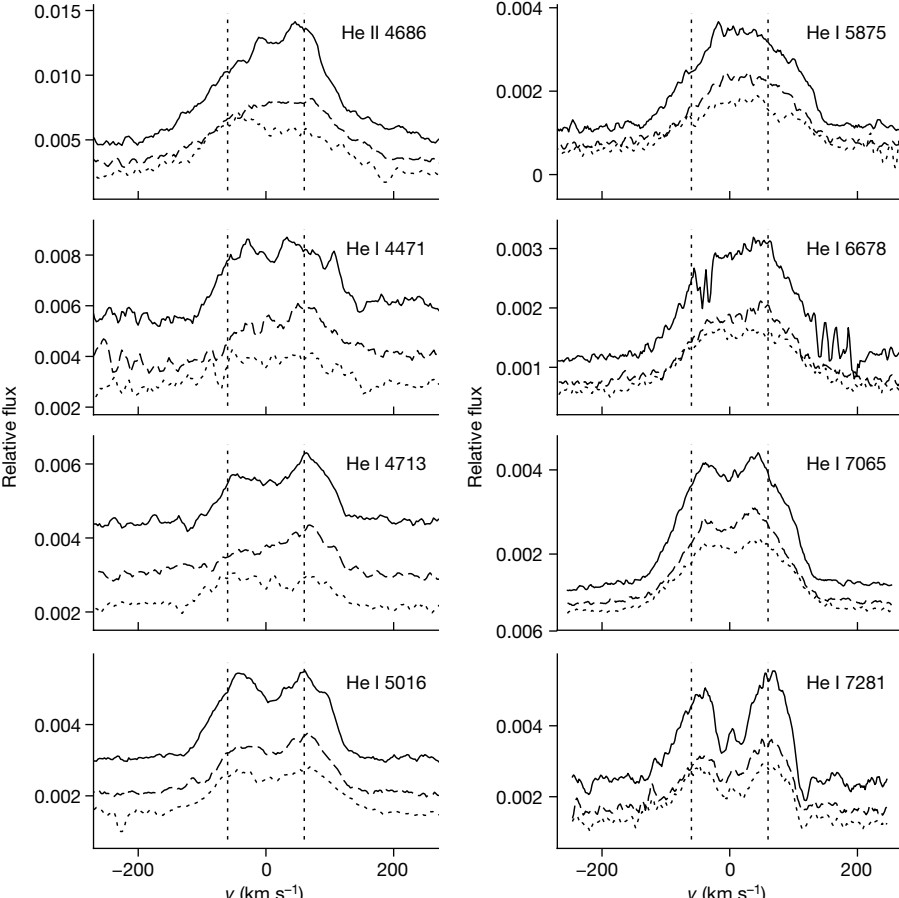

**Fig. 3 | Double-peak shape of optical emission lines.** Flux-normalized optical spectra of different He lines (as labelled on the top right of each panel) taken with the SALT/HRS spectrograph at three different epochs: 15 September 2020 (dashed lines), 5 October 2020 (dotted lines) and 6 October 2020 (solid lines).

The peak separation of all the main lines is similar, including that of He II. The vertical dotted lines indicate a peak separation of ±60 km s⁻¹. The relative variation of blue/red peaks is usually explained as the orbiting hotspot created by the impact of the accretion stream on the outer edge of the accretion disk.

ten or more, on timescales that depend on various parameters[24,25]. Even lower accretion rates result in explosive helium burning.

While the measured X-ray temperature is exactly in the range expected for steady He-shell burning, our measured luminosity is about ten times smaller than expected for accretion at the canonical rate. At the same time, the historical X-ray light curve, from Einstein (1979) and EXOSAT limits (1984–1986) to the ROSAT detection in 1992, and the XMM-Newton and eROSITA detections since 2019, suggests that the luminosity of [HP99] 159 is stable to within a factor of five (relative to the XMM-Newton value) for nearly 50 years (Extended Data Fig. 5). This indicates the possibility that helium accretion at rates well below the canonical one (that is, approximately $10^{-6} M_\odot$ year⁻¹) can still lead to stable helium burning.

Stable burning at low accretion rates has been suggested for the case that the accreting white dwarf is rapidly rotating[24,26]. In corresponding models, stable burning is found[21] down to $5 \times 10^{-7} M_\odot$ year⁻¹, and even for $3 \times 10^{-7} M_\odot$ year⁻¹ when allowing for fluctuations of the burning rate of a factor of three. In the latter situation, the X-ray luminosity at any given time may be up to a factor of three smaller, or larger, than the value deduced from a given accretion rate, assuming strictly stationary burning. If [HP99] 159 was near a luminosity minimum at present, which is more probable than it being near a maximum, its helium accretion rate could indeed be as high as $3 \times 10^{-7} M_\odot$ year⁻¹. Although we cannot exclude that the burning rate of [HP99] 159 is oscillating with a growing amplitude, leading to instability, the expected short timescale of the evolution renders this unlikely.

A lower than the canonical burning rate is consistent with our optical spectra. If the accretion rate in [HP99] 159 was substantially higher, a

wind from the white dwarf is expected[27]. This would manifest itself with emission lines, broadened by the wind velocity (on the order of thousands of km s⁻¹). Such broadened lines are not detected.

We have the following constraints on the mass of the He star: for initial He-star masses above about $1 M_\odot$, long-term stable evolution has been found[9]. The maximum possible initial mass depends on the assumptions concerning the wind. The present mass could be smaller than that. A rough upper bound on the present mass could be derived using the constraint that its luminosity is obviously smaller than that of the accretion disk. A He-star luminosity below about $1,000 L_\odot$ implies[28] that the current mass of the helium star is smaller than around $2 M_\odot$.

An orbital period of (1.16 days) 2.32 days suggests that the He star fills its Roche lobe radius of about (3) $4 R_\odot$, being a factor of about 10 larger than on the He main sequence. In this picture, as long as the mass of the He-star donor is larger than that of the white-dwarf accretor, mass transfer will proceed on the thermal timescale (approximately $10^5-10^6$ years), reducing the separation of the stars. Indeed, for He stars in the 0.8–2.0 $M_\odot$ range (corresponding to initial masses on the main sequence of 4–8 $M_\odot$), this thermal-timescale mass transfer[29] (during their subgiant or giant phases) is predicted to reach rates on the order of $10^{-7}-10^{-5} M_\odot$ year⁻¹, allowing for stable He burning. After mass-ratio inversion, the mass-transfer rate decreases and the binary widens. This may lead to the weak He-shell flash regime, consistent with [HP99] 159.

Various scenarios of white dwarfs accreting matter from a helium-star companion have been suggested to lead to Type Ia supernovae. At the lowest accretion rates, helium can pile up on the white dwarf and lead to a sub-Chandrasekhar-mass explosion after a critical amount of

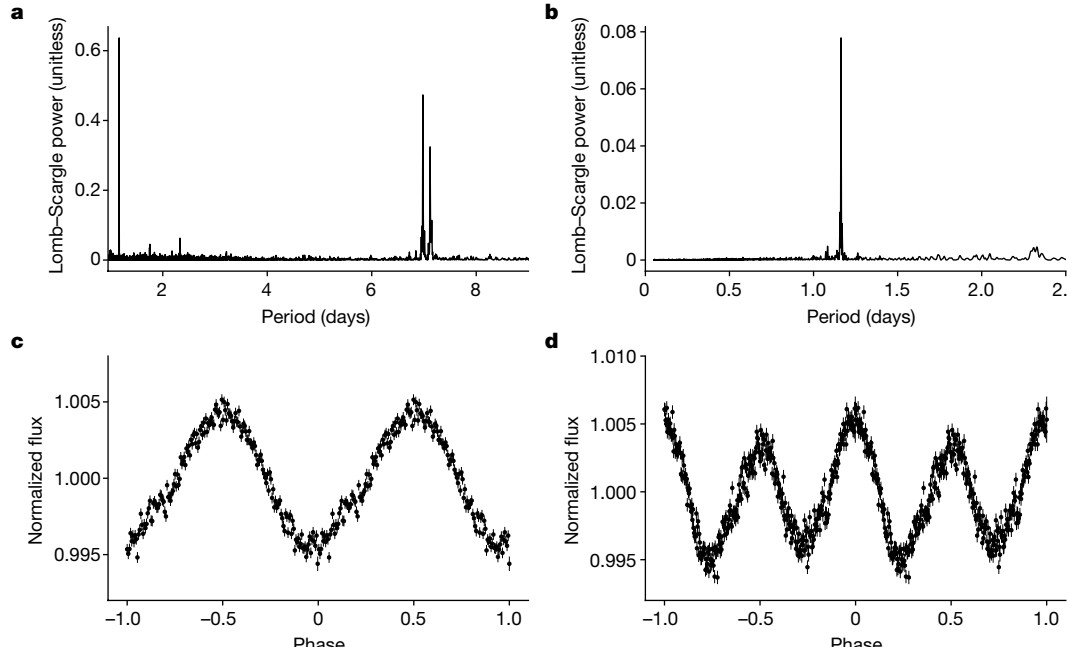

**Fig. 4 | Optical light variation. a**, **b**, Lomb–Scargle periodograms derived from the OGLE I-band data (**a**) and the TESS data (**b**). For the interpretation of the OGLE peaks, see Extended Data Table 1. **c**, **d**, The TESS data are folded with a period of $P = 1.1635$ days (**c**) and $P = 2.327$ days (**d**), corresponding to an ephemeris of 2,459,036.2885297858(14) + 1.1635*N or 2,459,036.288557(8) + 2.327*N (Barycentric Julian Date), respectively. The error bars for a given point represent

the root mean square error of the individual 10-min data points that went into that one bin. Although the Lomb–Scargle periodograms show the highest power at $P = 1.1635$ days, the folded light curve for the longer period (**d**) has a lower variance and a clear odd–even asymmetry. Lacking phase-resolved spectroscopy for a definite proof, we tentatively identify $P = 2.327$ days as the orbital period of [HP99] 159.

mass has been accumulated. However, in [HP99] 159, the X-ray emission implies continuous burning of the accreted matter and, consequently, a continuous growth of the white-dwarf mass. For this case, it has been suggested that the white dwarf undergoes a Type Ia supernova explosion once the Chandrasekhar mass is reached. A standard Type Ia explosion may strip 2–5% of the mass of the helium star[30], of which no signature has been observed so far. However, it has been suggested that Chandrasekhar-mass white dwarfs may undergo sub-energetic deflagrations[31], leading to sub-luminous so-called Type Iax supernovae, which are expected to strip off about ten times less mass from their helium donors[32]. Weak helium lines have been observed in the spectra of two Type Iax supernovae[13], and a helium donor star has been proposed for the Type Iax SN 2012Z based on deep pre-explosion imaging[33]. The recent detection of helium in the bright Type Ia SN 2020eyj (ref. [34]) indicates that helium donors may also sometimes trigger energetic white-dwarf explosions.

Although we do not know whether [HP99] 159 will evolve into a Type Ia supernova, its properties provide evidence for the pathway towards Chandrasekhar-mass explosions based on helium accretion being wider than previously thought. Its X-ray luminosity of about 1,800 $L_\odot$ corresponds to a stationary helium accretion rate of $1.5 \times 10^{-7} M_\odot$ year$^{-1}$, for which many models predict unstable burning at present[26]. However, [HP99] 159 seems to be relatively stable within the past 50 years. Stable burning for lower accretion rates, as perhaps enabled by rapid rotation[21], may allow lower-mass donors to push their companion white dwarfs to the Chandrasekhar mass. This may allow us to recover the SN Iax population within this scenario, which makes up about 30% of all Type Ia supernovae[13]. Folding our constraint on the radius of the white dwarf in [HP99] 159 with a white dwarf mass–radius relation[35], we find a current white dwarf mass of $1.20^{+0.18}_{-0.40} M_\odot$, implying that [HP99] 159 could undergo a Type Iax supernova explosion in the future.

When we assume that roughly 10% of all Type Ia supernovae in our Galaxy (around $10^{-3}$ per year (ref. [9])) follow the path of helium accretion leading to Type Iax explosions, and adopting a lifetime of $3 \times 10^5$ years (assuming that 0.3 $M_\odot$ needs to be transferred at $10^{-6} M_\odot$ year$^{-1}$), we

predict about 30 helium-accreting supersoft X-ray sources in the Milky Way at present. Scaling with the star-formation rate would yield a handful of systems in the LMC. The detection and study of more of these sources will probably allow us to tighten the constraints on the single-degenerate progenitor channel for Type Ia supernovae.

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

## Methods

### Optical photometry

**SkyMapper.** The optical brightness, measured by SkyMapper[36] (not simultaneously) is $g' = 15.82 \pm 0.02$ mag, $r' = 16.04 \pm 0.02$ mag, $i' = 16.41 \pm 0.01$ mag, $z' = 16.59 \pm 0.04$ mag and, after correcting for the Galactic and LMC reddening of $E_{B-V} = 0.105$ mag (see below), results in an absolute V-band magnitude of $M_V = -2.8$ mag (assuming a LMC distance[16] of 50 kpc). This is about 5 mag (or a factor of $2.5^5 = 100$) brighter than typical disks in high-accretion-rate nova-like cataclysmic variables [37] and still 15–40 times brighter for a face-on disk.

**OGLE.** The region of our X-ray source was monitored regularly in the V and I bands with the Optical Gravitational Lensing Experiment (OGLE)[38,39] at a cadence of 1–3 days. Photometric calibration is done by means of zero-point measurements in photometric nights and colour terms have been used for both filters when transforming to the standard V–I system. The long-term light curve during the period 2010–2020 shows variations by a factor of 1.3 and little colour variation (see Extended Data Fig. 3). A Lomb–Scargle periodogram identifies a period of $P = 1.1635$ days with the largest power (Fig. 4a,b), in agreement with $P = 1.163471$ days listed in the EROS-2 catalogue of LMC periodic variables (EROS-ID lm0454n2690)[40]. Two other strong peaks at longer periods are aliases (see Extended Data Table 1). A much smaller peak is seen at 2.327 days (see the paragraphs on TESS below).

**MACHO.** The source was also covered by the MACHO project[41], which monitored the brightnesses of 60 million stars in the Large and Small Magellanic Clouds and the Galactic bulge between 1992 and 1999. A visual (4,500–6,300 Å) and a red filter (6,300–7,600 Å) were used, the magnitudes of which were transformed to the standard Kron–Cousins V and R system, respectively, using previously determined colour terms[42].

**TESS.** The Transiting Exoplanet Survey Satellite[43] (TESS) is an all-sky transit survey to detect Earth-sized planets orbiting nearby M dwarfs. It continuously observes a given region of the sky for at least 27 days. For sources down to white-light magnitudes of about 16 mag, TESS achieves approximately 1% photometric precision in single 10-min exposures. However, its large plate scale (21″ pixel⁻¹) means that care must be taken with respect to blended sources.

[HP99] 159 was observed during all of TESS Sectors 27–39 (except Sector 33), that is, from July 2020 to June 2021. The analysis of [HP99] 159 is complicated by a 13-mag star at 12″ distance. Yet, the 1.16-day period found in OGLE data (which resolves these two stars) is clearly visible in a Lomb–Scargle periodogram of the TESS data (Fig. 4) as the strongest peak by far. There is a signal at 2.3268 days, exactly twice that of the OGLE period, at a significance of $3\sigma$. Although this is marginal, the folded (and rebinned) light curve shows a clear odd–even effect with smaller variance that leads us to believe that this is the true period, and the strong peak at 1.16 days is probably the first harmonic of this period. The small amplitude difference, at the 0.2% level, would explain that this is only marginally seen in the TESS periodogram. This period is also seen in the OGLE periodogram, demonstrating that it is a real feature. The phenomenon of asymmetrical maxima and minima, known in some detached binaries[44], is unique in interacting binaries and is especially puzzling given our inferred near-face-on geometry.

With the TESS light curve[45], we also carried out an independent, more sensitive search at even shorter periods that are inaccessible to OGLE. The TESS light curve was pre-whitened of the 1.16-day period and 25 of its harmonics, and the Fourier transform of the 'cleaned' data was calculated. There are no indications for a shorter period down to about 3 h (Fig. 4). There is also no signal at 0.538 days. This would be the fundamental period if the 1.16-day period were still an alias with the 1–3 days observing cadence of OGLE. On the other hand, two more periodicities are found, at $P_1 = 2.635$ h and $P_2 = 1.32$ h, with significances at the $4\sigma$ level (We assume that the noise is Gaussian and calculate the standard deviation in a 1,500-bin window ($\pm0.1$ cycles per day in frequency) around any identified peaks.). Given the non-Poissonian nature of the light curve after pre-whitening, we do not consider these two periods, which are not related harmonically, to be substantial enough for further investigation.

**Swift/UVOT.** A 1,061-s Swift observation was obtained on 9 August 2022, starting at 23:15 UT. Although not detected in X-rays (as expected, Extended Data Fig. 5), we detect [HP99] 159 in all filters of the Ultraviolet and Optical Telescope (UVOT), at AB magnitudes as follows: UVW2 = 15.29 ± 0.04 mag, UVW1 = 15.33 ± 0.04 mag, U = 15.44 ± 0.04 mag, B = 15.73 ± 0.04 mag and V = 15.93 ± 0.05 mag, for which the error is the quadratic sum of the statistical and systematic error. When added to the (non-simultaneous) measurements on the longer-wavelength bands (Extended Data Fig. 1), the spectral energy distribution is still well described by a straight power law, extending from 0.2 to 8.0 μm, without any sign of the He donor.

**Spectral energy distribution modelling and extinction correction.** The recent reddening map[46] of the LMC returns a much smaller reddening than previous estimates. Furthermore, it provides a combined reddening value for the Galactic foreground and the median LMC-intrinsic value, together with a spread owing to variation within the LMC. Instead of trying an arbitrary extinction correction, we instead forward-fold a power-law model to all the photometry from Swift/UVOT, SkyMapper, 2MASS and Spitzer. We fit for the power-law slope extinguished by a combination of Milky Way and LMC dust. The power-law model fit is very good and does not require a more complicated spectral model (Extended Data Fig. 1). The best-fit values are a power-law slope of $\nu^{1.48\pm0.02}$ and $E_{B-V}$ values of $0.01 \pm 0.01$ mag for the Milky Way and $0.14 \pm 0.01$ mag for LMC dust. The latter is larger than the $E_{B-V} = 0.11$ mag provided by the LMC reddening map[46] (composed of $E_{I-V} = 0.08$ mag to the centre of the LMC and a further $E_{I-V} = 0.06$ mag towards the far end of the LMC). More importantly, the slope of the spectral energy distribution is different from that expected for a standard accretion disk $F_\nu \propto \nu^{1/3}$ (Extended Data Fig. 1). This is very similar to the spectral energy distributions of other supersoft X-ray sources, such as CAL 83 (ref. [47]). The flatter slope has been interpreted as resulting from reprocessing of the high-luminosity soft X-rays, making the emission about 100–1,000 times larger than the accretion luminosity[48].

### Optical spectroscopy

Optical spectroscopy of our source was undertaken on the SALT. On 14 August 2020, a 1,200-s long-slit exposure was obtained using the RSS[49] in the 4,070–7,100-Å range (Fig. 2). Three further exposures (16 September 2020, 6 October 2020 and 7 October 2020), using the HRS[50], covered the 3,700–5,500-Å and 5,500–8,900-Å wavelength ranges. The primary reduction, which includes overscan correction, bias subtraction and gain correction, were carried out with the SALT science pipeline[51].

### X-ray analysis

**XMM-Newton.** 4XMM J052015.1−654426 was covered serendipitously in a 29-ks XMM-Newton observation (ObsId 0841320101, principal investigator Pierre Maggi) on 16/17 September 2019. The EPIC instruments were operating in full-frame mode, with thin and medium filters for the pn and MOS detectors, respectively. We used the XMM-Newton data analysis software SAS version 20.0.0 to process these data. Good time intervals were identified following the method described at https://www.cosmos.esa.int/web/xmm-newton/sas-thread-epic-filterbackground. A whole field-of-view light curve for single-pixel events with $10,000 < PI < 12,000$ is created and visually inspected for periods of flaring. A quiescent rate of less than 0.46 counts s⁻¹ is determined and a GTI file satisfying this condition is created and used to filter the observation. After this filtering and given the off-axis position (8.7 arcmin) of [HP99] 159, its resulting vignetted exposure was about 11.5 ks. The events used for the spectral analysis were filtered with the following expression using the SAS task evselect: '(PATTERN == 0) && (PI in [150 : 15000]) && (FLAG == 0)'. The SAS task especget was used

to extract (source and background) events from a circular region with radius 60″ centred on the position RA (2000.0) = 05 h 20 min 15.4 s, dec. (2000.0) = −65° 44′ 32″, as well as to calculate the response matrix file (RMF) and ancillary response file (ARF) for these events. The same was done with a circular region with radius 110″ centred on the position RA (2000.0) = 05 h 20 m 15.5 s, dec. (2000.0) = −65° 41′ 11″, to be used as the background only, after excising two point sources in that region. To estimate the spectral parameters of the source, a Bayesian approach was implemented using 3ML (refs. [52,53]). The analysis was restricted to the 0.2–2.3-keV energy band. The background and source contribution to the detected photons were modelled and folded through the appropriate responses to calculate posterior distributions of the spectral parameters. The source was modelled as an absorbed blackbody, using the 3ML models TbAbs*Blackbody (no separate abundances are used for the foreground Galactic and the LMC-intrinsic absorption). The background was modelled as a combination of instrumental background (read noise and fluorescence lines) and astrophysical background (Fig. 4) as follows: (1) a Gaussian line with normalization, line energy and width left free to account for the low-energy noise introduced by the readout electronics, (2) a Gaussian line with line energy and width fixed representing the Al-K fluorescent line near 1.5 keV, which is excited by particles in the camera body, (3) an unabsorbed APEC model with temperature left free to vary around 0.11 keV, accounting for the hot gas of the local bubble, (4) an APEC model with temperature allowed to vary around 0.22 keV absorbed by the average Galactic hydrogen column in the direction of the source, describing the contribution from the Galactic halo, and (5) a power law with a fixed slope of −1.41, absorbed by the combined hydrogen column of the Galaxy and the LMC in the direction of the source, arising from unresolved active galactic nuclei. The contribution of the particle background is negligible in our spectral range. The photons in the source extraction region were modelled by adding the source spectrum and the background spectrum, scaled by the ratio of the extraction areas. During the fit of the data, the parameters describing the background models were linked. We obtain the following best-fit values (errors at the $1\sigma$ level): $kT = 45 \pm 3$ eV, $N_H = (2.7 \pm 0.4) \times 10^{21}$ cm$^{-2}$ and an unabsorbed bolometric luminosity of $6.8^{+7.0}_{-3.5} \times 10^{36}$ erg s$^{-1}$; see Fig. 1. This implies an emission radius of $3,700^{+3,900}_{-1,900}$ kmkm, consistent with a white dwarf radius.

Apart from the possibility of the flux oscillations owing to the accretion rate being slightly below the burning rate, two other factors may contribute to the discrepancy of the measured versus expected X-ray luminosity. First, owing to the accretion of pure helium, the burning proceeds by means of the triple-$\alpha$ process[54], with $\log T(K) \approx 8.4$ and $\rho \approx 1,000$ g cm$^{-3}$ at the burning depth, leading to higher amounts of carbon and oxygen. Convective envelope mixing and subsequent wind ejection of CO-rich matter could lead to noticeable local X-ray absorption in the emission volume. Second, non-LTE model atmospheres (as frequently used for the supersoft phase in post-nova) usually give a higher peak intensity[55] than blackbody models (at the same temperature). Both effects, if taken into account in future work with improved data, would probably result in a higher X-ray luminosity (and white dwarf radius) than that estimated above.

**eROSITA.** [HP99] 159 = eRASSU J052015.3-654429 was detected by eROSITA[56] in each of the survey scans. Until the end of 2021, eROSITA scanned the source during five epochs as summarized in Extended Data Table 2. The X-ray position was determined from the combined four eRASS surveys to be RA (2000.0) = 05 h 20 min 15.52 s, dec. (2000.0) = −65° 44′ 28.9″ with a $1\sigma$ statistical uncertainty of 0.6″. The positional error is usually dominated by systematic uncertainties[57], which amount to 5″ in pointed and 1″ in scanning observations at present.

Owing to the unprecedented energy resolution (about 56 eV at 0.28 keV), eROSITA data are particularly sensitive to temperature changes of the source. Thus, we decided to perform spectral fitting despite the low number of counts. The spectral analysis was carried out using the five detectors with the on-chip aluminium filter (telescope modules 1, 2, 3, 4 and 6), avoiding the light leak in the other two detectors[56]. The eSASS[57] users version 211214 was used to process the data. Only single-pixel events without any rejection or information flag set were selected, using the eSASS task evtool. With the eSASS task srctool, a circular source region with radius 100″, centred on the coordinates RA (2000.0) = 05 h 20 min 16.6 s, dec. (2000.0) = −65° 44′ 27″ was defined to select source events. A background region of the same size and shape centred on RA (2000.0) = 05 h 21 min 9.4 s, dec. (2000.0) = −65° 46′ 0″ was defined, so as to lie at the same ecliptic longitude as the source region and hence in the scanning direction of eROSITA. The corresponding ARF and RMF files were created by the same eSASS task. Spectra were constructed by combining all events within the respective regions for each of the five epochs of observation. An absorbed blackbody was fitted to each of the spectra using 3ML. The priors of the free parameters were chosen on the basis of the XMM-Newton fit results. For the absorbing column, a Gaussian centred at $\mu = 2.7 \times 10^{21}$ cm$^{-2}$ and with a width of $\sigma = 0.4 \times 10^{21}$ cm$^{-2}$ was used. The prior on $kT$ was a Gaussian with $\mu = 45$ eV and $\sigma = 4$ eV, truncated at zero, and the prior on the normalization was a log-normal distribution with $\mu = \log(400)$ and $\sigma = 1$. For the eROSITA data, the background was not modelled because of the low number of counts; rather, the data were binned to have at least one background photon in every bin and a profile Poisson likelihood was used. For the five epochs, we obtain best-fit temperatures of $kT_1 = 42^{+3}_{-2}$ eV, $kT_2 = 44^{+3}_{-2}$ eV, $kT_3 = 42^{+3}_{-2}$ eV, $kT_4 = 42 \pm 2$ eV and $kT_5 = 43 \pm 2$ eV. The corresponding fluxes are listed in Extended Data Table 2 and shown in Extended Data Fig. 5, together with the fluxes (or limits) of the other X-ray missions.

**ROSAT.** [HP99] 159 was originally identified[14] in a 8.3-ks ROSAT/PSPC pointed observation (ID 500053p) of April 1992. We have reanalysed this observation and find the source with a vignetting-corrected count rate of $0.005 \pm 0.001$ PSPC counts s$^{-1}$ (40 ± 8 source counts). A blackbody fit with free parameters leads to $kT = 38 \pm 15$ eV, $N_H = (0.9^{+3.2}_{-0.3}) \times 10^{21}$ cm$^{-2}$ and an unabsorbed bolometric luminosity of $1.3^{+41.7}_{-1.0} \times 10^{36}$ erg s$^{-1}$. A fit with a fixed, XMM-derived temperature of 45 eV is statistically indistinguishable (owing to the very small number of counts and the low energy resolution) and results in an absorption-corrected bolometric luminosity of $1.7^{+41}_{-1.0} \times 10^{36}$ erg s$^{-1}$, consistent within the errors of the free fit. A fit with fixed, XMM-derived temperature and $N_H$ is substantially worse.

[HP99] 159 was not detected during the ROSAT all-sky survey, with a PSPC count rate upper limit of <0.012 counts s$^{-1}$. Using the best-fit spectral model of the above ROSAT pointed observation leads to a luminosity limit <$2.5 \times 10^{36}$ erg s$^{-1}$, whereas using the XMM-derived spectral parameters leads to <$3.2 \times 10^{37}$ erg s$^{-1}$. For consistency with the Einstein and EXOSAT upper limits, we choose to plot the latter value in Extended Data Fig. 5.

## Arguments against an AM CVn interpretation

The He-dominated accretion disk and the N II and Si II lines (Extended Data Fig. 2) allow the possibility of an AM CVn nature of [HP99] 159. However, a number of reasons argue against this interpretation. (1) AM CVn objects have luminosities[58] in the range $10^{30}$–$10^{32}$ erg s$^{-1}$. For this to be applicable to [HP99] 159, it would need to be at a distance of order 100 pc. (2) This is incompatible with the Gaia data, which suggest a minimum distance of 8–12 kpc. (3) Similarly, all AM CVn stars have large proper motion[58], on the order of 0.5″ year$^{-1}$, owing to their vicinity. This is a factor 100 larger than that of [HP99] 159. (4) Finally, and most convincing, the velocity shift of all the strong lines clearly indicates LMC membership. At that distance, an AM CVn system is incompatible with the parameters we observe.

## Comparison with known similar systems

To our knowledge, the only other 'known' system of this kind was the progenitor of the He nova V445 Pup[59]. A pre-outburst luminosity of

$\log(L/L_\odot) = 4.34 \pm 0.36$ would be compatible with a $1.2$–$1.3\,M_\odot$ star burning helium in a shell[60]. No optical spectrum exists of the progenitor; the post-outburst spectra are H-deficient, with the strongest lines being C II and Fe II (ref. [61]). On the basis of photographic plates taken before the outburst, an optical modulation by a factor of 1.25 and a period of 0.650654(10) days was found and interpreted as orbital variation of a common-envelope binary[62]. There are three possibilities for the X-ray non-detection: (1) the flux oscillations during burning with phases of low luminosity[25], (2) the substantial Galactic foreground absorption in the case that the X-ray spectrum was as similarly soft as [HP99] 159 or (3) an only slightly lower temperature as compared with [HP99] 159, which would shift the emission below the X-ray detection window. Thus, the progenitor of the He nova V445 Pup could have been an object similar to [HP99] 159.

## Data availability

Data from the ROSAT, XMM-Newton, Swift and TESS missions, as well as from the OGLE and MACHO projects, are publicly available. eROSITA data of the first survey (eRASS1) of [HP99] 159 will be made public as part of the eRASS1 data release, scheduled for April 2023 at present. Data of the subsequent eROSITA surveys (eRASS2 and later) will be made public according to the plan as provided at https://erosita.mpe.mpg. de/erass/. The optical spectra taken with the SALT telescope are available at https://cloudcape.saao.ac.za/index.php/s/g8M1q1ya8ef7Fzd.

## Code availability

All data analysis code is publicly available, as referenced in the text.

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

**Acknowledgements** R.W. is supported by the German Research Foundation (DFG) under contract GR1350/17-1 and J.M.B. by the DFG-funded Collaborative Research Center SFB 1258. We thank M. Freyberg (MPE Garching) for discussion of the XMM-Newton background modelling and S. Rappaport (MIT) on the TESS time-series analysis. J.G. is grateful to the Swift team for the rapid scheduling of the UVOT observation. This work is based on data from eROSITA, the soft X-ray instrument aboard SRG, a joint Russian–German science mission supported by the Russian Space Agency (Roskosmos), in the interests of the Russian Academy of Sciences represented by its Space Research Institute (IKI) and the Deutsches Zentrum für Luft- und Raumfahrt (DLR). The SRG spacecraft was built by Lavochkin Association (NPOL) and its subcontractors and is operated by NPOL with support from the Max Planck Institute for Extraterrestrial Physics (MPE). The development and construction of the eROSITA X-ray instrument was led by the MPE, with contributions from the Dr. Karl Remeis Observatory Bamberg & ECAP (FAU Erlangen-Nürnberg), the University of Hamburg Observatory, the Leibniz Institute for Astrophysics Potsdam (AIP) and the Institute for Astronomy and Astrophysics of the University of Tübingen, with the support of the DLR and the Max Planck Society. The Argelander Institute for Astronomy of the University of Bonn and the Ludwig-Maximilians-Universität München also participated in the science preparation for eROSITA. The eROSITA data shown here were processed using the eSASS software system developed by the German eROSITA Consortium. This paper uses public-domain data obtained by the MACHO Project, jointly financed by the US Department of Energy through the University of California, Lawrence Livermore National Laboratory under contract no. W-7405-Eng-48, by the National Science Foundation through the Center for Particle Astrophysics of the University of California under cooperative agreement AST-8809616 and by the Mount Stromlo and Siding Spring Observatory, part of the Australian National University. Some of the observations reported in this paper were obtained with the SALT under programme 2018-2-LSP-001. This paper includes data collected by the TESS mission. Funding for the TESS mission is provided by NASA's Science Mission Directorate. Resources used in this work were provided by the NASA High-End Computing (HEC) Program through the NASA Advanced Supercomputing (NAS) Division at Ames Research Center for the production of the Science Processing Operations Center (SPOC) data products.

**Author contributions** C.M., F.H., R.W. and P.M. analysed the X-ray data, with J.M.B. providing the 3ML environment. D.A.H.B. and I.M.M. obtained the SALT spectra, A.U. provided the OGLE data and R.J. and R.V. analysed the TESS data. J.G. recognized the He burning nature and, with N.L., H.R., J.B. and K.W., derived the binary system constraints. All authors contributed to the scientific discussion and the writing of the manuscript.

**Funding** Open access funding provided by Max Planck Society.

**Competing interests** The authors declare no competing interests.

**Additional information**
**Correspondence and requests for materials** should be addressed to J. Greiner.

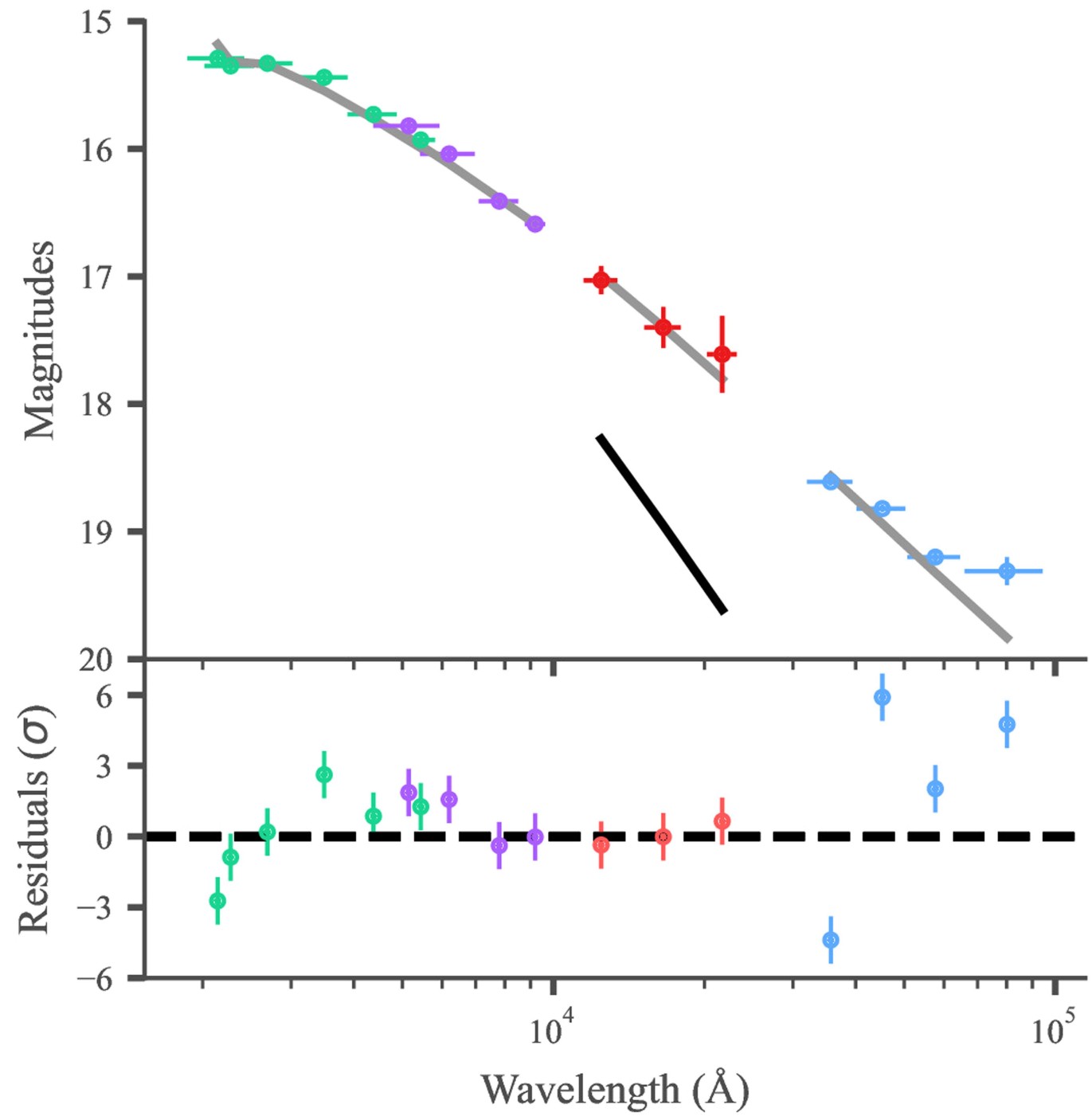

**Extended Data Fig. 1 | Spectral energy distribution of [HP99] 159.** The collection of non-simultaneous photometry of [HP99] 159 from SkyMapper (griz; purple), 2MASS (JHK; red) and Spitzer (IRAC-1 to IRAC-4; blue), together with near-simultaneous five-filter ultraviolet–optical photometry with Swift/UVOT (green). Error bars are at the 1σ level. The extinction has been fit by forward-folding a power law and allowing for the sum of Milky Way and LMC dust extinction curves (best-fit $E_{B-V} = 0.14 \pm 0.01$ mag). The orbital variability amplitude of ±0.15 mag is the probable reason for the scatter in the non-simultaneous data but has not been included as 'systematic' error in the fitting. The slope of the observed spectral energy distribution (grey line; $F_\nu \propto \nu^{1.48\pm0.02}$) is clearly different from that expected from a non-irradiated accretion disk (black line: $F_\nu \propto \nu^{1/3}$), consistent with the properties of canonical supersoft X-ray sources[48].

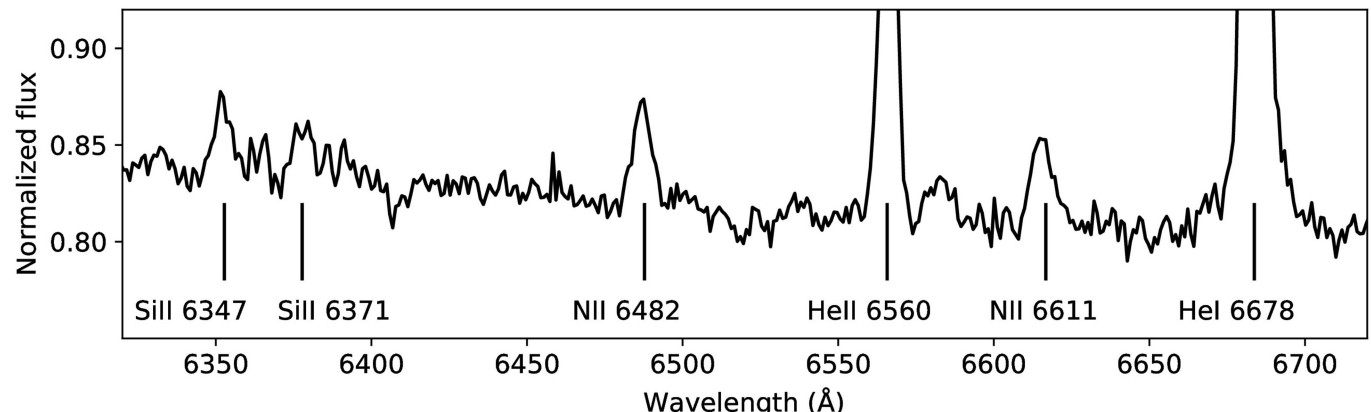

**Extended Data Fig. 2 | Faint nitrogen and silicon lines.** Zoom-in of the low-resolution SALT spectrum, showing four of the seven N II and Si II emission lines identified in addition to the He lines.

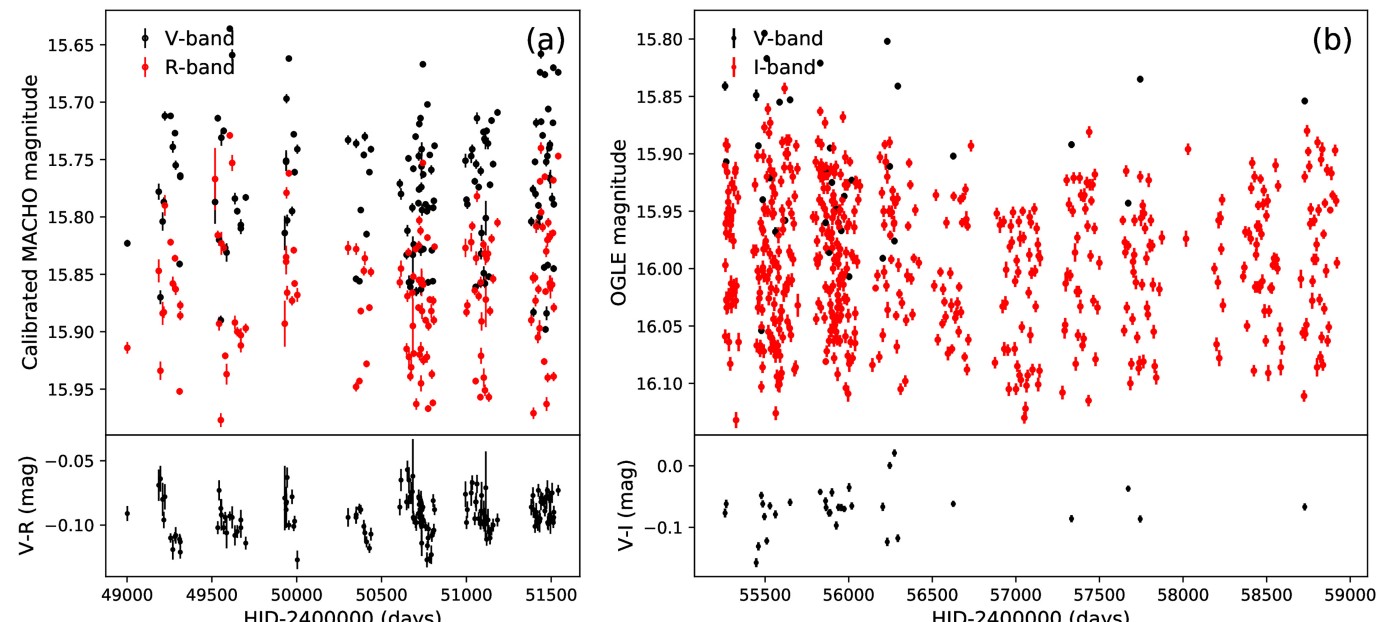

**Extended Data Fig. 3 | Optical long-term light curve of [HP99] 159.** MACHO (panel **a**; January 1993 to January 2000) and OGLE (panel **b**; March 2010 to March 2020) light curves of [HP99] 159. Error bars are at the 1σ level. The MACHO colour has a further systematic error of ±0.028 mag. The V-magnitude offset between the two panels is a result of the accuracy of the absolute photometric calibration of the MACHO data.

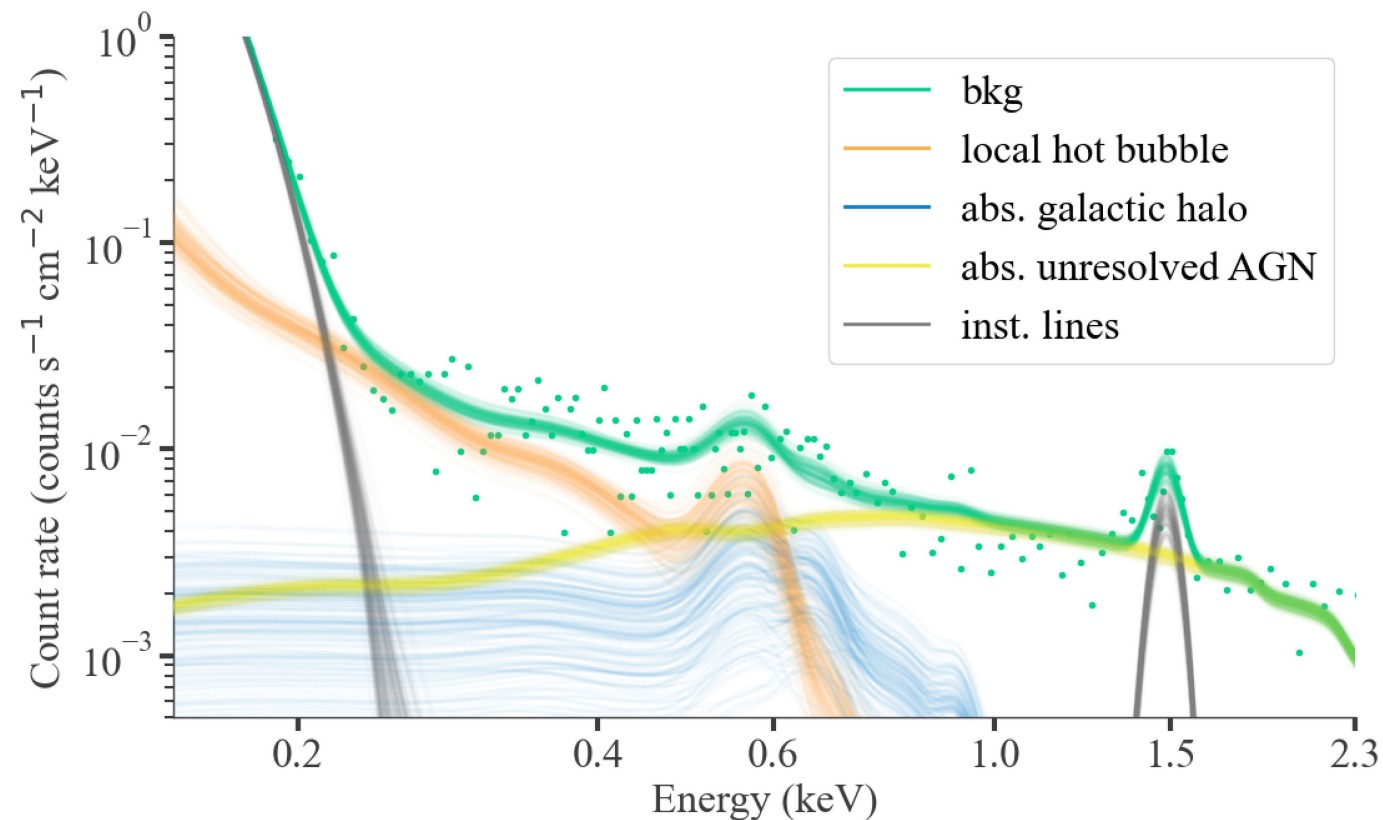

**Extended Data Fig. 4 | XMM-Newton X-ray background modelling.** The XMM-Newton X-ray background is extracted from a distinct region on the detector but located close to the source region and, in a first step, separately modelled as the sum of four components, as labelled. During the spectral fit of the photons from the source extraction region, the background spectrum is scaled by the ratio of the extraction areas and the parameters of three of the four background model components were linked; for the read-noise component (steeply falling grey component below 0.3 keV), because of its strong variation over the detector, the parameters were left free.

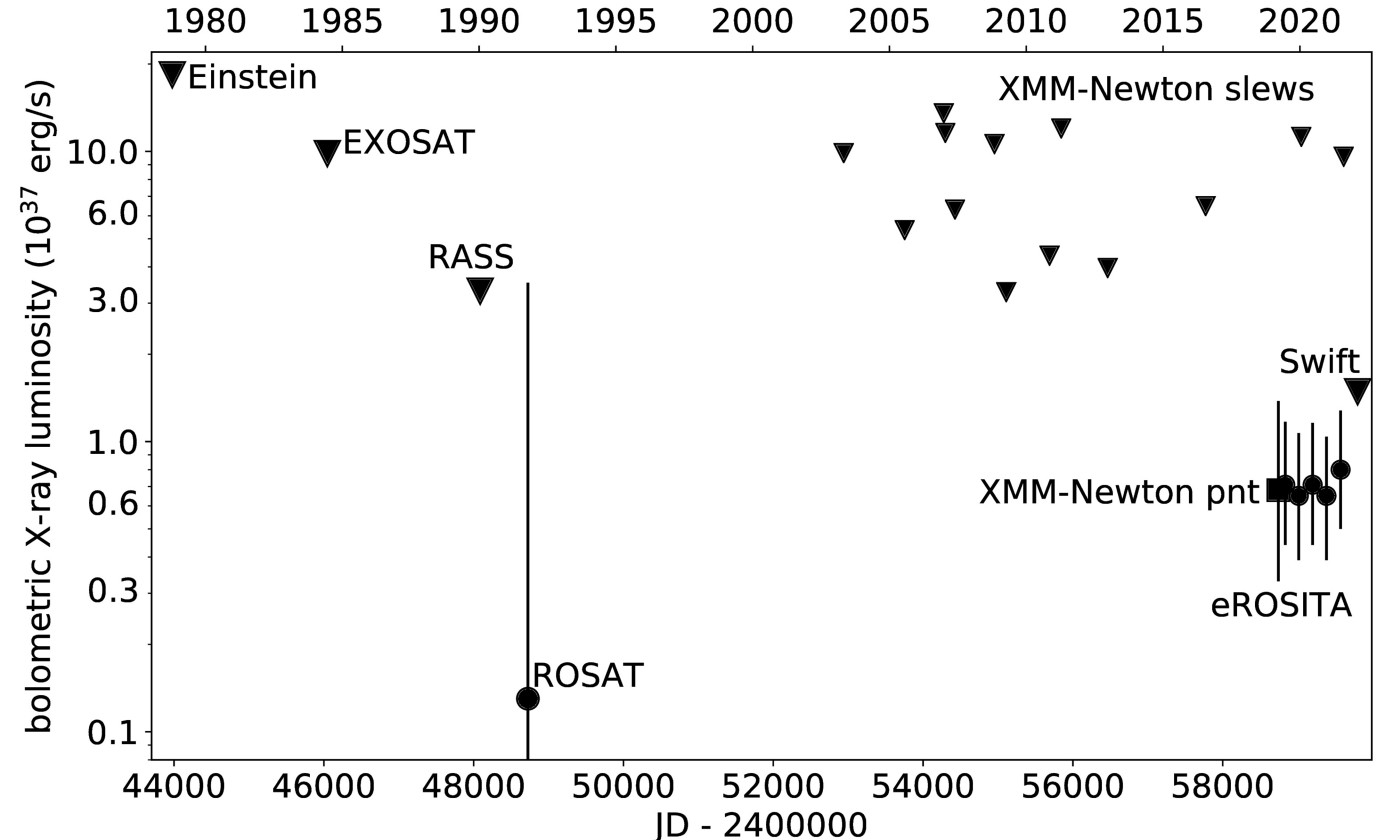

**Extended Data Fig. 5 | Long-term X-ray light curve.** Compilation of X-ray $2\sigma$ upper limits (triangles) and detections (filled circles with $1\sigma$ error bars) from previous X-ray missions. The upper limits from Einstein, EXOSAT and the XMM-Newton slews have been computed using http://xmmuls.esac.esa.int/ upperlimitserver/ with $kT = 60$ eV and $N_H = 10^{21}$ cm$^{-2}$ and then transformed to the best-fit XMM-Newton spectral values using WebPIMMS (https://heasarc.gsfc.nasa.gov/cgi-bin/Tools/w3pimms/w3pimms.pl).

**Extended Data Table 1 | Peaks in the Lomb–Scargle periodogram of the OGLE data**

| Peak No. | Maximum power | P (days) | f (1/days) | 1.0 + f (1/days) | 1.0 - f (1/days) | f - $f_{III}$ (1/days) |
|:---:|:---:|:---:|:---:|:---:|:---:|:---:|
| I | 0.6367 | 1.1635 | 0.8595 | 1.8595 | 0.1405 | 0.7190 |
| II | 0.4733 | 6.9794 | 0.1433 | 1.1433 | 0.8567 | $2.77 \times 10^{-3}$ |
| III | 0.3271 | 7.1171 | 0.1405 | 1.1405 | 0.8595 | – |

Notes: (i) $1/(f_{II}-f_{III})$=360.8 days indicates that these two frequencies are 1-year aliases of each other. (ii) $1-f_{III}$=0.859494≈$f_I$, suggesting that $f_I$ and $f_{III}$ are 1-day aliases. (iii) Possible shorter periods are: $1/(1+f_{III})$=0.87680 days or $1/(1+f_I)$=0.53778 days.

**Extended Data Table 2 | X-ray observations of [HP99]159**

| Mission | observation time $T_{start} - T_{stop}$ (UTC) | exposure (ks) | count rate[a] (cts s$^{-1}$) | luminosity[a] (erg s$^{-1}$) |
|---|---|---|---|---|
| *Einstein* | 1979-04-12 17:10–1979-04-12 17:50 | 2.4 | <0.01 | <$1.8 \times 10^{38}$ |
| EXOSAT | 1986-01-21 04:17–1986-01-21 12:00 | 27.8 | < 0.0047 | <$9.8 \times 10^{37}$ |
| ROSAT Survey | 1990-07-12 01:33–1990-07-16 02:54 | 2.4 | <0.012 | <$3.2 \times 10^{37}$ |
| ROSAT 500053p | 1992-04-09 16:10–1992-04-13 19:15 | 8.3 | 0.005±0.001 | $1.7^{+41}_{-1.0} \times 10^{36}$ |
| XMM-Newton | 2019-09-16 18:35–2019-09-17 03:25 | 28.6 | 0.056±0.003 | $6.8^{+7.0}_{-3.5} \times 10^{36}$ |
| eRASS0/1[b] | 2019-12-11 23:33–2019-12-27 15:33 | 1.2[c] | 0.10±0.01[d] | $7.1^{+4.5}_{-2.7} \times 10^{36}$ |
| eRASS1/2[b] | 2020-06-07 00:57–2020-06-23 20:57 | 1.0[c] | 0.10±0.01[d] | $6.5^{+4.2}_{-2.6} \times 10^{36}$ |
| eRASS2/3[b] | 2020-12-13 14:33–2020-12-28 14:33 | 0.9[c] | 0.06±0.01[d] | $7.1^{+4.5}_{-2.7} \times 10^{36}$ |
| eRASS3/4[b] | 2021-06-11 13:57–2021-06-27 05:57 | 1.1[c] | 0.09±0.01[d] | $6.5^{+3.9}_{-2.6} \times 10^{36}$ |
| eRASS4/5[b] | 2021-12-17 15:33–2021-12-19 15:33 | 0.1[c] | 0.17±0.08[d] | $8.0^{+4.8}_{-3.0} \times 10^{36}$ |

[a]Count rate is in the 0.2–2-keV band and luminosity is foreground-absorption corrected. [b]eROSITA test scans from 8 December 2019 to 11 December 2019 are designated as eRASS0. The source position was covered by these test scans and in the early phase of eRASS1. The approximately 2-week visibility of [HP99]159 for eROSITA starts at the end of each formal eRASS survey and extends into the start of the following eRASS: we group this into 'epochs' of continuous coverage. [c]Net exposure per telescope after correcting for vignetting, averaged over the five telescope modules used. [d]Net source count rate after correcting for vignetting, summed over the five telescope modules used.