## [Peer Review File · Nature]

Manuscript Title: A helium-burning white dwarf binary as a supersoft X-ray source

Reviewer Comments & Author Rebuttals

Reviewer Reports on the Initial Version:

Referees' comments:

Referee #1 (Remarks to the Author):

Based on a thorough analysis of observations with X-ray instruments partially dating back to the 1990s in combination with new optical observations, the authors propose a model for the supersoft X-ray source [HP99] 159. The key point of the manuscript is this novel and original interpretation rather than new detections and observational results. I will therefore comment on this aspect of the manuscript and not on technical details of the observations.

Two questions are important in this context: i) Are the interpretation of the data and the proposed model of for the observed system convincing? ii) What are the implications of identifying such a system?

Regarding i), the authors present an impressive wealth of data. Based on ROSAT, XMM-Newton and eROSITA data, they identify [HP99] 159 as a supersoft X-ray source and follow the standard interpretation of a white dwarf burning accreted material steadily at its surface. Optical spectra taken with SALT are presented. They are dominated by helium features and show some nitrogen and silicon lines. Therefore the authors propose that the object is a system in which a white dwarf accretes helium-rich material from a companion and at a rate that allows for steady burning. The companion is suggested to be a helium star. In the Methods part, the authors make a number of points why the source is not likely to be an AM CVn object. They mainly refer to the attributed membership of the object to the LMC and the thus implied luminosity. The high-resolution SALT spectra indicate an accretion disk at a low inclination angle dominating the optical emission. In my view, these arguments are convincing. Here, however, one has to note that the explanation of the observed luminosity in the framework of the model proposed by the authors is not completely unproblematic as discussed in the main text. The other uncertainty is that of the orbital period of the system. A period of 2.327 days is assumed but the associated feature in the Lomb-Scargle periodogram is rather inconspicuous. It is argued that the folded light curve shows a smaller variance for this period -- is this sufficient to prefer this value? As the authors say, phase-resolved spectroscopy is needed to settle the question. For both uncertainties, however, the implications for the modeling are discussed and they are no fundamental concern for the general interpretation. The emerging model for [HP99] 159 is that of a massive white dwarf accreting material from a helium star which is steadily burned at the surface. This interpretation seems convincing to me.

In my view, point ii) is critical for a publication in Nature. What do we learn from the observations and the interpretation in terms of the model proposed by the authors? The main point here seems to be the mere fact of the existence of the binary system in the inferred configuration. Confusingly,

the authors follow two lines of arguments. The first is the claim that we know such systems must exist; and now we have finally found an example. This is reasoned by the observations of a "large number of hydrogen-free supernovae with small ejecta masses". I guess this refers to Type Ia supernovae (if so, why is such a strange denomination used? To circumvent the potential problem of ejecta contamination with helium?). For Type Ia supernovae, however, the progenitors are rather unclear. Different channels and explosion scenarios are discussed and a system with a He star as a donor to the exploding white dwarf is not the only -- and probably also not the most common -- possibility, as also stated by the authors later in the manuscript when they estimate the number of expected systems. Another argument is that such systems could form sources of gravitational waves and of ionizing flux in cosmic reionization. For all effects, other scenarios are discussed. Systems such as the one observed may contribute, but seems unlikely that they are dominant. I therefore find this line of arguments little compelling. In my view, it even makes the manuscript weaker.

The reverse logic is more convincing: As the authors state in their second line of arguments, finding such a system supports the idea of this class of objects indeed contributing to the listed effects. But this should be further substantiated for the four areas mentioned (or at least some of them). At the moment, the points made seem rather generic and the relation to the observed system parameters does not become immediately clear.

1) Type Ia supernovae: What would be needed to constrain the mass of the accreting white dwarf? It is said in the manuscript (p. 12) that it can easily be grown to the Chandrasekhar limit -- can this claim be quantified? Is the system observed likely to be a progenitor of a Type Ia supernova? What about the helium potentially stripped off from the companion when the white dwarf explodes and polluting the ejecta? Such processes have been simulated (e.g. Liu et al., *ApJ* 774, 37, 2013). Can the mass of He stripped by the supernova be estimated for the studied system? Is it in agreement with observations of Type Ia supernova? Could the observed system be a progenitor of a Type Iax supernova rather than a normal Type Ia? How does it compare to the observations of the progenitor system of SN 2012Z (McCully et al., *Nature* 512, 54)?

2) Bridging the gap between sdO/sdB and WR stars: It is argued that the observation of systems such as presented in the manuscript allows for constraints on wind parameters. What exactly is needed here? Can anything be deduced from the existing observations of [HP99] 159?

3) Type Ibn or Type Ibc supernova, or a merger of compact objects: To make the text better readable for the general audience, it could be stated here that it is now the fate of the He star that is discussed (as opposed to an explosion of the white dwarf in 1). Some constraints are given, but how do they compare to the properties of the studied system? The Ibc supernova channel seems to require higher He star masses than that derived for the system. For too low masses of the He star, the fate discussed is a merger of two white dwarfs. The total mass of the merged object will exceed the Chandrasekhar limit and it is argued that this causes a Type Ia supernova. The currently discussed mechanisms for Type Ia supernovae resulting from white dwarf mergers, however, are not necessarily related to the Chandrasekhar mass limit.

4) Contribution to cosmic reionization. If systems such as studied here are important, the interesting point is perhaps the additional contribution from the radiation originating from steady He burning. How would this change the picture?

Overall, I think that point i) is convincingly discussed in the manuscript. For publication in *Nature*, point ii) is critical. The arguments made in this context should be sharpened to support the claim of the importance of the observational finding. I think that there is sufficient potential for this, so that a

publication in Nature could be considered after revision.

typo:

Methods part, p27: (iv) Finally, and most convincing, the velocity shift of all 582 the strong lines clearly indicate LMC membership.

-> indicates

Referee #2 (Remarks to the Author):

This paper reports on the observation of a hydrogen-free supersoft X-ray source. This is a binary star system in which a helium-rich, hydrogen-free donor star is transferring mass to a white dwarf companion through an accretion disc. At the surface of the white dwarf the helium burns to carbon and beyond generating the soft X-rays. The importance of this find lies in the nature of some type-Ia supernova progenitors in which a helium star companion transfers mass to a white dwarf. The burning of the helium as it accretes prevents any form of nova eruption that might blow off the accreted material. In this way the white dwarf can reach a mass of 1.38 solar masses when carbon ignites at its centre setting off the supernova. Though predicted theoretically and in binary star population synthesis studies no firm candidate has been identified observationally. This system remedies that situation and paves the way for future observations that can eventually lead to a quantitative estimate of the number of type-Ia supernovae that are produced in this way. As with all compact binary stars there are implications for future gravitational wave detections. The high temperatures of naked helium stars make them important contributors to interstellar ionization and again the opportunity to quantify the contribution of binary stars to such ionizing sources is extremely important.

As a theoretician I cannot comment on many of the details of the observations but the work appears sound and convincing. I have a few, relatively minor but important comments on the manuscript. Most importantly the references seem to be a bit haphazard. Often the actual originator of an idea is ignored with the reference being made to a recent study instead. A thorough examination of the sources is advised so that the original authors are referred to. At line 186 the statement that wind parameters for sdO and WR stars are well constrained needs a reference. At line 68 you state that simulations with NEBULAR are for mixed hydrogen helium gas. This is fine for a ratio of hydrogen to helium of one to ten but how do things change if there is no hydrogen at all?

And there are some problems with the presentation. The symbol ' \sim ' meaning 'is asymptotically equal to' is a verb and should not be used as an adjective. Sadly such misuse is now common in astronomical journals but this does not make it correct. Quantities plotted in figures are dimensionless so the figure labels must reflect this by dividing by the appropriate physical unit. For example a velocity label might be written as $v/\text{km s}^{-1}$ in TeX. Avoid the future tense when the present is called for. Make sure that in subclauses introduced by "which", "which" actually refers to what you intend.

Often it is better to begin a new sentence with "This" instead. "First" is itself an adverb so "firstly" has a superfluous ending.

Christopher Tout

Reply to Referee comments

Referee #1:

> In my view, point ii) is critical for a publication in Nature. What do we learn from the observations and the interpretation in terms of the model proposed by the authors? The main point here seems to be the mere fact of the existence of the binary system in the inferred configuration. Confusingly, the authors follow two lines of arguments. The first is the claim that we know such systems must exist; and now we have finally found an example. This is reasoned by the observations of a "large number of hydrogen-free supernovae with small ejecta masses". I guess this refers to Type Ia supernovae (if so, why is such a strange denomination used? To circumvent the potential problem of ejecta contamination with helium?).

REPLY: The "hydrogen-free supernovae with small ejecta masses" related to the SNe Ibc; we had erroneously placed the reference for SN Ia at the end of that sentence: our apologies for the confusion.

> For Type Ia supernovae, however, the progenitors are rather unclear. Different channels and explosion scenarios are discussed and a system with a He star as a donor to the exploding white dwarf is not the only -- and probably also not the most common -- possibility, as also stated by the authors later in the manuscript when they estimate to number of expected systems. Another argument is that such systems could form sources of gravitational waves and of ionizing flux in cosmic reionization. For all effects, other scenarios are discussed. Systems such as the one observed may contribute, but seems unlikely that they are dominant. I therefore find this line of arguments little compelling. In my view, it even makes the manuscript weaker.

REPLY: We follow the advice and concentrate on the implications for the SN Ia progenitor question.

> 1) Type Ia supernovae: What would be needed to constrain the mass of the accreting white dwarf? It is said in the manuscript (p. 12) that it can easily be grown to the Chandrasekhar limit -- can this claim be quantified?

REPLY: Our determination of the WD radius results in a mass close to the Chandrasekhar limit, although still with rather large error bar. We quantify this now in the text (l. 153).

> Is the system observed likely to be a progenitor of a Type Ia supernova?

REPLY: We discuss this now in some detail (lines 144-154). The most likely outcome is indeed a Type Iax SN.

> What about the helium potentially stripped off from the companion when the white dwarf explodes and polluting the ejecta? Such processes have been simulated (e.g. Liu et al., ApJ 774, 37, 2013). Can the mass of He stripped by the supernova be estimated for the studied system? Is it in agreement with observations of Type Ia supernova?

REPLY: We are grateful this question is brought up. We discuss it in detail (lines 129-143). It strengthens the arguments towards Type Iax SNe.

> Could the observed system be a progenitor of a Type Iax supernova rather than a normal Type Ia? How does it compare to the observations of the progenitor system of SN 2012Z (McCully et al., Nature 512, 54)?

REPLY: A SN Iax is a valid option, as it supposedly strips off much less Helium from the donor. We have added text to that effect. HP99 is less luminous than SN 2012Z (and V445 Pup), but also notably bluer, so it is located above the He-donor strip as shown in Fig. 2 of McCully, consistent with our earlier statements of being brighter than the He star.

> 2) Bridging the gap between sd0/sdB and WR stars: It is argued that the observation of systems such as presented in the manuscript allows for constraints on wind parameters. What exactly is needed here? Can anything be deduced from the existing observations of [HP99] 159?

REPLY: We have now removed this implication from the discussion.

> 3) Type Ibn or Type Ibc supernova, or a merger of compact objects: To make the text better readable for the general audience, it could be stated here that it is now the fate of the He star that is discussed (as opposed to an explosion of the white dwarf in 1). Some constraints are given, but how do they compare to the properties of the studied system? The Ibc supernova channel seems to require higher He star masses than that derived for the system. For too low masses of the He star, the fate discussed is a merger of two white dwarfs. The total mass of the merged object will exceed the Chandrasekhar limit and it is argued that this causes a Type Ia supernova. The currently discussed mechanisms for Type Ia supernovae resulting from white dwarf mergers, however, are not necessarily related to the Chandrasekhar mass limit.

REPLY: As this part relied perhaps too much on speculations, we removed this implication from the discussion.

> 4) Contribution to cosmic reionization. If systems such as studied here are important, the interesting point is perhaps the additional contribution from the radiation originating from steady He burning. How would this change the picture?

REPLY: In lack of a quantitative prediction, we also removed this implication from the discussion.

> Overall, I think that point i) is convincingly discussed in the manuscript. For publication in Nature, point ii) is critical. The arguments made in this context should be sharpened to support the claim of the importance of the observational finding.

REPLY: Following the advice of Referee #1, we concentrated on the most important aspect, which is the implication for Type Ia SNe, which is now discussed quantitatively and in some detail. Accordingly, we have removed the other three potential implications in order to sharpen the focus (and shorten the text)

> typo: Methods part, p27: (iv) Finally, and most convincing, the velocity shift of all the strong lines clearly indicate LMC membership: -> indicates

REPLY: corrected

Referee #2:

> ...I have a few,
> relatively minor but important comments on the manuscript. Most
> importantly the references seem to be a bit haphazard. Often the
> actual originator of an idea is ignored with the reference being made
> to a recent study instead. A thorough examination of the sources is
> advised so that the original authors are refereed to.

REPLY: We have carefully re-examined the references, and for the (now main) topic of He-burning and supersoft X-ray sources we are confident that we have not missed relevant original literature. Should we still miss an important reference, we would be grateful for the advise of Referee #2.

> At line 186 the statement that wind parameters for sd0 and WR stars
> are well constrained needs a reference.

REPLY: since we removed the discussion of the mass bridging, this has not been included

> At line 68 you state that simulations
> with NEBULAR are for mixed hydrogen helium gas. This is fine for a
> ratio of hydrogen to helium of one to ten but how do things change is
> there is no hydrogen at all?

REPLY: This was meant to provide a rough indication of the conditions, but is it well appreciated that NEBULAR is not a tool to derive proper physical conditions in a gas (as e.g., CLOUDY), not even talking about the conditions in a strongly radiated accretion disk. We therefore deleted this sentence.

> The symbol \sim ;
> meaning \sim is asymptotically equal to \sim ; is a verb and should not be
> used as an adjective. Sadly such misuse is now common in astronomical
> journals but this does not make it correct.

REPLY: we have replaced it by the more appropriate \approx symbol

> Quantities plotted in
> figures are dimensionless so the figure labels must reflect this by
> dividing by the appropriate physical unit. For example a velocity
> label might be written as $v/\text{km s}^{-1}$ in TeX.

REPLY: We had used the convention of "(...)" meaning 'in units of' which is equivalent to the division by the unit. Looking through recent Nature volumes, we see that our usage dominates. However, if editorial policies prefer the unit division style, we are happy to change this.

> Avoid the future tense when the present is called for.

REPLY: We tried our best.

> Make sure that in subclauses introduced by "which",
> "which" actually refers to what you intend.
> Often it is better to begin a new sentence with "This" instead.

REPLY: we have re-phrased a handful of sentences where the logic was broken

> "First" is itself an adverb so "firstly" has a superfluous ending.

REPLY: corrected

Reviewer Reports on the First Revision:

Referees' comments:

Referee #1 (Remarks to the Author):

As I stated in my previous report, the interpretation of the observations of the supersoft X-ray source [HP99] 159 by the authors seems convincing. My concern was the astrophysical implication that the observation of such a system might have. In revising the text, the authors have done a very good job of streamlining their arguments. I find their discussion strong but still balanced. It makes a clear point that the results have significant consequences for progenitor scenarios of thermonuclear supernovae. Therefore I recommend publication of the article in Nature.

There is only one typo I noticed:

line 106: "If [HP99]159 were currently be near a luminosity minimum..." Something seems to be wrong with this sentence -- should the "be" be deleted?

Referee #2 (Remarks to the Author):

I am happy that all points made have been adequately addressed.